# Nanofibrous Hydrogel Nanocomposite Based on Strontium-Doped Bioglass Nanofibers for Bone Tissue Engineering Applications

**DOI:** 10.3390/biology11101472

**Published:** 2022-10-08

**Authors:** Soheila Zare, Mahnaz Mohammadpour, Zhila Izadi, Samaneh Ghazanfari, Samad Nadri, Hadi Samadian

**Affiliations:** 1Student Research Committee, Zanjan University of Medical Sciences, Zanjan 45154, Iran; 2Department of Chemistry, Faculty of Sciences, Tarbiat Modares University, Tehran 1411713116, Iran; 3Pharmaceutical Sciences Research Center, Health Institute, Kermanshah University of Medical Sciences, Kermanshah P.O. Box 671551616, Iran; 4USERN Office, Kermanshah University of Medical Sciences, Kermanshah P.O. Box 671551616, Iran; 5Aachen-Maastricht Institute for Biobased Materials (AMIBM), Faculty of Science and Engineering, Maastricht University, 6167 RD Geleen, The Netherlands; 6Department of Biohybrid and Medical Textiles (BioTex), AME-Helmholtz Institute for Biomedical Engineering, RWTH Aachen University, 52072 Aachen, Germany; 7Zanjan Pharmaceutical Nanotechnology Research Center, Zanjan University of Medical Sciences, Zanjan 45154, Iran; 8Zanjan Metabolic Diseases Research Center, Zanjan University of Medical Sciences, Zanjan 45154, Iran; 9Research Center for Molecular Medicine, Hamadan University of Medical Sciences, Hamadan 6517838736, Iran

**Keywords:** bone tissue engineering, hydrogel, nanocomposite, bioglass nanofibers, strontium

## Abstract

**Simple Summary:**

Currently, bone defects, diseases, and injuries are common and global problems. These defects can be treated with several surgical methods and bone grafting, but these methods have limitations, including immune disorders, risk of infection, long-term recovery, movement problems, and high costs. A promising treatment option for bone replacement is the design and construction of scaffolds that mimic the properties of bone tissue and provide a suitable environment for cell and tissue growth. Achieving successful results in this method is dependent on the composition and structure of materials used as scaffolds. Bone is a composite consisting of a mineral fraction, mainly a combination of calcium phosphate, and an organic matrix. Here, we designed and produced a porous, non-toxic, and degradable scaffold made of alginate natural polymer and bioactive glass that contains strontium as well as the common elements of bioglasses—silica, calcium, sodium, and phosphorus. The scaffold is degraded at an optimized rate with the simultaneous proliferation and growth of cells, thus providing a suitable environment for the growth and development of new tissue and blood vessels. The outcomes of this study presented this scaffold as a functional structure to be used in treating bone defects and reconstructing damaged bone.

**Abstract:**

The main aim of the current study is to fabricate an osteocompatible, bioactive, porous, and degradable bone tissue engineering scaffold. For this purpose, bioactive glasses (BGs) were chosen due to their similarity to bone’s natural mineral composition, and the effect of replacing Ca ions with Sr on their properties were considered. First, strontium-containing BGs (Sr-BGs) were synthesized using the electrospinning technique and assembled by the sol–gel method, then they were incorporated into the alginate (Alg) matrix. Photographs of the scanning electron microscope (SEM) showed that the BG nanofibers have a diameter of 220 ± 36 nm, which was smaller than the precursor nanofibers (275 ± 66 nm). The scaffolds possess a porous internal microstructure (230–330 nm pore size) with interconnected pores. We demonstrated that the scaffolds could be degraded in the acetate sodium buffer and phosphate-buffered saline. The osteoactivity of the scaffolds was confirmed via visual inspection of the SEM illustrations after seven days of immersing them in the SBF solution. In vitro assessments disclosed that the produced Alg-based composites including Sr-BGs (Alg/Sr-BGs) are blood-compatible and biocompatible. Accumulating evidence shows that Alg/Sr-BG (5%, 10%, and 15%) hydrogels could be a promising scaffold for bone regeneration.

## 1. Introduction

Bone is a connective tissue that can regenerate and self-heal. Still, some bone defects, such as osteoporotic and large defects, can interfere with constructional stability and normal biomechanics of bone or lead to its deformities [1,2]. In most cases, these defects can be treated with multiple surgical methods and bone grafts, often associated with restrictions such as long-term healing, immune disorders, risk of infection, movement problems, and high pain. They also significantly impact health and social care [1,2,3]. The efficiency of commonly used methods is low, and often repetitive surgical procedures are required. There is no guarantee of complete regeneration of the bone defect and the surrounding soft tissue, either [2,3]. Currently, alternative therapies to surgical methods and bone grafts to augment the reconstruction of bony defects have been studied, wherein bone tissue engineering (BTE) and each of its key components, namely stem cells, scaffold biomaterials, and growth factors, have received a lot of attention [4]. An interesting branch of this research is the design and fabrication of bioactive scaffolds to support or direct bone cell attachment, proliferation, migration, and differentiation via providing a proper mechanical and chemical environment; this leads to the vascularization and formation of new tissues [3,4].

A suitable scaffold should mimic the fine structures, mechanical strength, and biological, chemical, and electrical properties of bone tissue; it should be biocompatible and biodegradable as well [4,5,6,7,8]. Bone is a porous composite material consisting of both inorganic and organic components in which the inorganic phase is mainly nanocrystalline hydroxyapatite [4,6,7,8]. It should be noted that the scaffold must accommodate primary mineralization and, therefore, requires a large surface area, high interconnected porosity, and proper degradation rate. Having these characteristics, the scaffold will have the essential requirements for cell growth, tissue formation, growing tissue vascularization, and replacement of new tissue [4,6,7,8].

In the present study, sodium alginate (Alg)—a natural ionic polysaccharide—was selected as the scaffold matrix because of its nontoxicity, biodegradability, cytocompatibility, cost-effectiveness, and in situ gelation ability [4,9]. The carboxyl groups of α-L-guluronic acid units in the Alg polymer chain have a high affinity for ionic crosslinking with definite divalent metal ions, such as Ca^2+^ and Sr^2+^, which leads to in situ gelation [9,10]. This unique feature has made it possible to use Alg in wound-dressing applications, tissue engineering scaffolds, and drug release applications. Previous research has proven the ability of calcium alginate gel to be used as a scaffold or cell carrier for bone reconstruction [11,12]. However, alginate alone does not have an optimal degradation rate, proper bioactivity behavior, or potential for angiogenesis and healing of lesions, and does not provide sufficient load-bearing properties for orthopedics applications. Therefore, numerous strategies were proposed to develop more functional Alg composites, such as incorporation of inorganic materials including hydroxyapatite and BG into Alg [4,9,10,13,14].

BGs (a combination of glass and ceramic) are a bioactive inorganic material having similar components to the natural inorganic portion of bone. Studies have demonstrated that BGs are desirable for hard tissue regeneration applications and possess multiple bioactivities and excellent osteoconductivity ability. They showed angiogenesis and osteogenesis properties both in vitro and in vivo via releasing their ions, especially silica and calcium. Further, many reports indicated that osteoblast proliferation and the different behavior and biological performance of BGs were promoted by the addition of some therapeutic metal ions such as Sr, Zn, Mg, and Cu [13,15,16,17].

Among the dopant metal ions, Sr stimulates bone healing and growth processes and reduces bone resorption. Sr usually functions similarly to Ca and can simply substitute it. This is the only element in the bone that is associated with bone compression strength. With the inspiration of this knowledge, a few efforts have been made into incorporating Sr into bioactive glasses using various methods such as sol–gel. To this end, the Sr-substituted BG was a better biomaterial than 45S5 Bioglass^®^ and induced more apatite formation [16,18,19,20].

In the current work, the electrospinning of BG in combination with the sol–gel technique was used to create continuous fibers with diameters ranging from submicron to nanoscale. The electrospun bioglass fibers reportedly have a high degradation rate, superior mechanical strength, and high surface area and accelerate bone-like apatite deposition when soaked in SBF. They also possess the potential for proliferation and differentiation of bone marrow-derived cells. The biodegradation rate of a scaffold is another crucial feature that must be considered in the design of bone substitutes and bone tissue engineering platforms. A fundamental aspect of this assessment is matching the intended scaffold degradation rate with the healing and regeneration rate of the damaged bone tissue [21,22,23]. In this study, we prepared an alginate-base bioactive hydrogel that was reinforced with Sr-doped bioglass nanocomposites, where undoped BG was used as a control. Simulated body fluid (SBF) was utilized to investigate the biomineralization potential of the scaffold, and its cytocompatibility and blood compatibility were tested on osteoblast-like cells (MG-63 cells) and human fresh anticoagulated blood, respectively. Since bone scaffolds are intended to be at the implant site for a long time, analysis of swelling properties, degradation behavior, and porosity rate of gel-composite scaffolds were necessarily accomplished.

## 2. Materials and Methods

### 2.1. Materials

Tetraethyl orthosilicate (TEOS; 99.99%), triethyl phosphate (TEP; 99.5%), calcium nitrate tetrahydrate (Ca(NO_3_)_2_. 4H2O; 99.60%), strontium nitrate tetrahydrate (Sr(NO_3_)_2_; hydrochloric acid (HCl), polyvinyl alcohol (PVA; Mol. wt. ∼ 125,000 g mol−1), and all used solvents were purchased from Sigma-Aldrich, St. Louis, MO, USA. Sodium alginate (SA) and calcium chloride crosslinker (CaCl_2_. 4H_2_O) were acquired from Merck (Darmstadt, Germany). An MTT assay kit was purchased from Roth (Karlsruhe, Germany). Fetal bovine serum (FBS), high-glucose Dulbecco’s modified Eagle’s medium (DMEM)/F-12 cell culture medium, trypsin–EDTA, and penicillin–streptomycin (Pen–Strep) were obtained from Gibco (Waltham, MA, USA). The MG-63 cell line was provided from the cell bank of the Iran Pasteur Institute, Tehran, Iran, and tissue culture plates were obtained from SPL, Pyeongtaek, Korea. All chemicals were used as received without any further purification, except those mentioned specifically.

### 2.2. Preparation of Bioglass Nanofibers and Sr-Doped Bioglass Nanofibers

#### 2.2.1. Fabrication of Precursor Solutions of Electrospinning

The present work employed the integration of sol–gel with the electrospinning technique to fabricate two types of bioglass nanofibers. In order to synthesize the undoped BG solution, 148 mg of Ca(NO_3_)_2_. 4H_2_O was fully dissolved in a solution containing EtOH (4 mL) and deionized H_2_O (2 mL). After adding 0.1 mL of HCl (1 M), 1.34 g of TEOS, and stirring for 30 min, 0.116 g of TEP was added to the solution. The resulting solution was stirred continuously for 24 h at room temperature (RT). The Sr-doped bioglass (Sr-BG) solution was obtained by a similar procedure in which the first 0.132 g of Sr(NO_3_)_2_ was dissolved in the EtOH/H_2_O solution, and other substances were introduced into the Sr(NO_3_)_2_ ethanolic solution in the same manner.

Before electrospinning, it was necessary to regulate the resulting silica sol rheology by adding a secondary solution—here, PVA solution. For this purpose, 0.8 g of PVA polymer was added to 5 mL of H_2_O and stirred at 50 °C for 3 h for polymer dissolution to occur completely. Next, two types of electrospinning precursor solutions, PVA/BG and PVA/Sr-BG were created by mixing the PVA solution and each of the bioglass solutions in a 1:1 weight ratio and stirring at 45 °C for 3 h.

#### 2.2.2. Electrospinning Process

An electrospinning machine (Electronics, FNM, Tehran, Iran) [24] comprising a high-voltage supply, spinneret system, syringe pump, and collector was used. All parameters were set to optimize the settings as follows: DC voltage: 20 kV, nozzle-to-collector distance: 12 cm, needle tip diameter: 18 gauge, and feeding rate: 0.5 mL h-1. The mentioned precursor solutions were electrospun on the aluminum foil-covered collector using the same parameters. Eventually, the electrospun mats were separated from the aluminum foil and then placed at 600 °C with heating at a rate of 4 °C min^−1^ for 5 h in a furnace aimed at removing PVA polymer and calcination.

### 2.3. Fabrication of 3D Nanocomposite Scaffolds

The synthesized BG nanofibers and Sr-BG were pulverized into fine particles; then, various amounts of Sr-BG nanoparticles were poured into deionized water to provide varying concentrations (5%, 10%, and 15% *w*/*w* (Sr-BG/Alg ratio)) and dispersed by vigorous stirring and bath sonication for 3 h. A proper amount of Alg was added to each solution so that its concentration in each solution reached 2% *w*/*v*. The mixtures were stirred at ambient temperature until the alginate was entirely dissolved and homogeneous hydrogel solutions were formed. The Alg-based hydrogels were crosslinked in 200 mM CaCl_2_ solution for 4 h, rinsed with deionized water three times to remove the excess ions, and finally transferred to an ultra-low-temperature freezer. After 24 h, frozen samples were freeze-dried under vacuum at −50 °C to fabricate hydrogels as 3D nanocomposite scaffolds.

### 2.4. Characterization Techniques

#### 2.4.1. Morphological Observation

The internal structures of Sr-BG nanofibers, before and after heat treatment, and lyophilized Alg/Sr-BG and Alg/BG scaffolds were characterized by SEM using a scanning electron microscope (Philips XL-30, Eindhoven, The Netherlands) at an accelerating voltage of 20 kV. The samples were cut along the longitudinal direction and cross sections of the slices were sprayed with a thin layer of gold using a sputter coater (SCD 004, Balzers, Germany). The electron microscope was also equipped with energy-dispersive X-ray spectroscopy (EDS) that was recruited for the elemental analysis of the surface of the calcined nanofibers and freeze-dried nanocomposite gels. The average diameter of nanofibers and mean pore size of 3D composites were quantified from SEM images using Image J (NIH, Bethesda, Rockville, MD, USA).

#### 2.4.2. Crystallinity Assessment

The fuzzy structure and the absence/presence of crystallization in the nanofibers were investigated after calcination using an X-ray diffraction analyzer (XRD, Philips XL-30, Germany, 40 kV, 30 mA) with Cu-Ka radiation (λ = 1.54056 Å). The 2θ range of 10–90° was adopted with a step size of 0.08°s^−1^ for the investigation.

#### 2.4.3. Specific Functional Group Identification

Spectra of three samples (BG, Alg, and Alg/BG) were probed to estimate the distinguished functional groups by utilizing an FTIR spectrometer (Spectrum RX1 FTIR system, PerkinElmer, Waltham, MA, USA). The analysis was recorded at RT in the wavenumber range of 400–4000 cm^−1^ using the KBr (spectroscopic grade) pellet technique.

#### 2.4.4. Swelling Kinetics

The scaffold’s ability for swelling in an aquatic environment was calculated according to Equation (1):(1)$$\mathrm{Swelling}\;\mathrm{ratio}\;=\frac{\;W_{0}-W}{W_{0}}\times 100$$

For this measurement, three equivalent segments of Alg/Sr-BG (5%, 10%, and 15% Sr-BG) and Alg/10% BG scaffolds in the dry state were weighed (*W*_0_) and then soaked in deionized water. The samples were collected from the solution at pre-set time intervals, carefully placed on a filter paper to remove the superficial water, and their weight was recorded as swollen weight (*W*).

#### 2.4.5. Weight Loss Measurement

The biodegradation % (*w*/*w*) was computed through the weight loss measurement of the Alg-based scaffolds containing BGs in two buffer solutions, PBS (pH 7.2) and acetate sodium/acetic acid (pH 4.2), at 37 °C for 60 days. Three specific amounts of freeze-dried scaffolds were carefully cut, weighted, registered as *W_i_*, and submerged in buffer solution. On days 7, 14, 30, and 60, the samples were taken out from the solution, entirely dried, and weighed again as *W_t_.* The degradation rate of the nanocomposite scaffolds was calculated using Equation (2):(2)$$\mathrm{Degradation}\;\mathrm{rate}\;\mathrm{of}\;\mathrm{weight}\;\mathrm{loss}\;\left(\%\right)=\frac{W_{t}-W_{i}}{W_{i}}\times 100\%$$

### 2.5. Biological Evaluations

#### 2.5.1. Osteoactivity Investigation

The osteoactivity evaluation was conducted using the simulated body fluid (SBF, pH 7.2). The SBF solution for the bioactivity test of Sr-BG and BG gels was prepared based on the literature [25,26]. The dried specimens were immersed in SBF and incubated at 37 °C, as reported by Kokubo et al. [27]. Seven days after immersion, the scaffolds were filtered, rinsed with distilled water, and lyophilized. The changes that had taken place on glass crosslinked hydrogels were viewed by SEM.

#### 2.5.2. Hemocompatibility Evaluation

The fresh blood was diluted with sterilized normal saline at a 4:5 ratio for hemolysis examinations of nanocomposite hydrogels and collected in anticoagulant tubes. Phosphate-buffered saline (PBS) and deionized water were employed as negative and positive controls, respectively. Each sample, Alg/Sr-BG (5%, 10%, and 15% Sr-BG) and Alg/10% BG scaffolds were incubated with 200 μL of blood in 1.5 mL Eppendorf (EP) tubes at 37 °C for 60 min. After completing this, each tube was centrifuged at 1500 rpm for 5 min and the absorbance of the withdrawn supernatant was read by a microplate reader (Anthos 2020, Biochrom, Berlin, Germany) at 545 nm. The ratio of hemolysis of samples, as a %, was obtained according to the formula as follows (Equation (3)):(3)$$\mathrm{Hemolysis}\;\left(\%\right)=\frac{D_{t}-D_{nc}}{D_{pc}-D_{nc}}\times 100\%$$
where *D_t_* is the absorbance of the sample, *D_nc_* is the absorbance of the negative control, and *D_pc_* is the absorbance of the positive control.

#### 2.5.3. Cell Viability Assessment

This assessment was performed to estimate the efficacy of crosslinked Alg containing Sr-doped BG at three different concentrations (5%, 10%, and 15% *w*/*w*) on the viability of the MG-63 osteosarcoma cell line, and results were compared with the case where calcium alginate with undoped BG was used as a scaffold. The selected scaffolds were plated in a 96-well polystyrene plate after sterilization. Triplicate wells were made for each individual sample. The MG-63 cells were cultured in DMEM containing 10% FBS and 1% penicillin/streptomycin for 24 h at 37 °C. The cells were harvested, and a cell suspension with the final concentration of 7 × 10^3^ cells/well was added to the surface of the scaffolds and incubated for 1 and 3 days in a CO_2_ incubator (37 °C, 5% CO_2_, and 90% relative humidity). During the cultivation period, the cells’ status was visually monitored by light microscopy, and each well’s medium was replaced with fresh DMEM medium every 24 h. An MTT assay was performed to quantify the proliferation. At the end of each treatment day (1 and 3 days), the culture medium was removed from the plate and the scaffolds comprising cells were rinsed with PBS, after which 200 µL of 5 mg/mL MTT solution was added to every well and incubated in the dark for 4 h at 37 °C and 5% CO_2_. The MTT solution was then replaced by 50 μL of DMSO to dissolve the purple formazan crystals that formed during incubation. Finally, the amount of light absorbed at 570 nm was read using a microplate reader (Anthos 2020, Biochrom, Berlin, Germany) and the cell viability % was calculated using Equation (4):(4)$$\;\;\;\;\;\mathrm{Cell}\;\mathrm{viability}\;\left(\%\right)=\frac{{\;\mathrm{Sample}}_{\mathrm{abs}}}{{\;\mathrm{Control}}_{\mathrm{abs}}}\times 100\%$$

Cells cultured with culture medium only and without the nanocomposite hydrogels were used as control and were considered 100% growth.

### 2.6. Statistical Analysis

Statistical Product and Service Solutions (SPSS) software pack version 23.0 (IBM Corp., Armonk, NY, USA) was used for data analyses. Results are introduced as mean and standard deviation and values of *p* < 0.05 were considered by using one-way analysis of variance (ANOVA). Comparison analyses between two or more groups were performed using Student’s *t*-test. All experiments were measured in triplicate and the average and error bar were obtained.

## 3. Results

### 3.1. Characterizations

#### 3.1.1. Morphology and Composition of Nanofibers and Nanocomposites

Using SEM images, the surface morphology of PVA/Sr-BG nanofibers before and after heating was analyzed, and the relevant results are illustrated in Figure 1. The PVA/Sr-BG nanofibers in Figure 1A are uninterrupted, smooth, and bead-free with a straight orientation, while in Figure 1B, a random orientation is presented for calcined Sr-BG nanofibers, and their fusion with each other is evident. The nanofibers in Figure 1A have an average diameter of approximately 275 ± 66 nm (calculated using Image J software), which is slightly larger than the heated form with a mean diameter of 220 ± 36 nm. According to the images in Figure 1, the PVA/Sr-BG fibers’ surface became coarse and uneven after heating at 600 °C and removing the organic polymer, indicating their conversion to BG nanofibers. EDS spectroscopy determined the Sr composition of bioactive glass nanofibers both before and after heating (Figure 1C,D), and the results showed no significant difference.

Figure 2 shows the SEM images of the cross-sectional area of Alg/BG, Alg/Sr-BG 5%, Alg/Sr-BG 10% and Alg/Sr-BG 15% nanofiber–hydrogel composites (A–D, respectively). These alginate-based scaffolds displayed a highly interconnected porous structure and inserted micrographs (Figure 2C,D) exhibited a homogeneous distribution of the nanofibers all over the hydrogel matrix. This can be attributed to using multifold ultrasonic steps in the production procedure. The Sr-free Alg/BG composite indicates an architecture with smoother surfaces, lower pore interconnectivity, and a larger average pore size (313.66 µm for Figure 2A relative to Sr-containing samples, 275.6, 235.72, and 228.60 µm for Figure 2B–D, respectively). It has been reported that pore size in the range from 75 μm to 250 μm is suitable for a scaffold of bone tissue engineering, which, according to the measurements, places composites with 10 and 15% Sr-BG into this category [28,29,30].

The elemental composition of the lyophilized hydrogels was evaluated by EDS spectroscopy, and the results of mapping scans in Figure 3 verified the presence of Si, Ca, P, and Sr elements in the samples, which coincides with the initial composition of the sol–gel mixture used to make glass nanofibers. Additionally, according to the EDS maps, the amount of Ca decreases in proportion to the increase in the amount of Sr, which shows the replacement of calcium with strontium.

#### 3.1.2. FTIR and XRD

Figure 4 shows the FTIR spectrum of freeze-dried Alg/BG aerogels with spectra of raw Alg and 600 °C heat-treated bioactive matter, and the most specific signals follow for each. In connection with Alg, peaks of stretching vibrations of OH^−^ groups, both asymmetric and symmetric vibrations of carboxylate (–COO^−^) salts and glycosidic C–O–C linkage, were located at 3395, 1614, 1415, and 1072 cm^−1^, respectively [4,31]. The FTIR spectrum of powdered silicate glasses revealed peaks at 796 and 1078 cm^−1^ that can be assigned to the symmetric and asymmetric stretching modes of the Si–O–Si bonds. The peaks at 1382 and 1475 cm^−1^ were related to the C–O bond stretching vibrations and the bending vibration of the O–H bond. Other than that, broad peaks located at 3415 cm^−1^ were triggered by the vibration of OH groups. The band at 1631 cm^−1^ was attributed to the vibration of hydroxyl groups of water molecules that were absorbed on the surface [32,33,34]. Following embedding the BG in the gel, FTIR spectra show the combined bands of Alg and BG with the changes in intensity and position of some of the peaks compared to pure Alg; this confirms that BGs were well-incorporated into the scaffold [34].

In Figure 5, the broad peaks with low intensity in XRD spectra of the calcined Sr-BG nanofibers indicate the amorphous nature and lack of crystalline phase in the sintered nanofibers similarly found in previous studies. In a bioactive glass network, control of ion release rate and bioactivity capacity of the material are affected by this amorphous nature [32,33,34].

#### 3.1.3. Swelling Kinetics

Swelling capacity, among the most fundamental properties of scaffolds, indicates the potential of a scaffold for the uptake and preservation of fluids in its skeleton. This feature increases the pore diameter and porosity of the scaffold and allows more substantial contact with the surrounding tissues, which are favorable for infiltration, attachment, and migration of cells. The swelling property directly correlates with the porosity [14,35,36]. When Alg/Sr-BG constructs and the Alg/BG control scaffold were immersed in an aqueous solution to study swelling for 4 days, a striking reverse correlation was seen between the Sr-BG nanoparticle concentration and the swelling degree of these hydrogel composites, as shown in Figure 6. The Alg/BG composite gel presented a swelling ratio alteration of less than 10% from 2 to 4 h; this means that the swelling degree of the gel attained equilibrium in about 4 h. In contrast, the 5%, 10%, and 15% Sr-BG-laden hydrogels, within the first 24 h, increasingly adsorbed solution, after which they saturated and the curve flattened. This may be due to dwindling pore size with an increase in Sr-BG powder content, as reported by morphological studies, or to the stronger interaction between alginate and Sr-BG [14,37,38].

#### 3.1.4. Biodegradation Profile

Insomuch that, in a logical timespan, the scaffold degradation potency can present enough space for neo-bone formation and vascular growth, in contrast, when the degradation rate is rapid, it leads to undesirable tissue defects in the implant site [39,40]. The biodegradation of fabricated gel composite substrates was tested in two different environments, PBS solution (pH 7.4) and AcOH buffer (pH 4.2), to mimic the natural physiological status and acidic condition of the bone fracture site, respectively. The degradation profile is shown in Figure 7, which depicts that the degradation rate of all composite scaffolds comprising bioglass in acetate buffer at pH 4.2 (Figure 7A) was more rapid than that in the PBS solution (Figure 7B). Additionally, the degradation rate of lyophilized gelling scaffold possessing Sr-substituted bioglass was higher in both circumstances compared to the Sr-free samples and was enhanced by increasing the amount of Sr-BG in the crosslinked hydrogel. This is likely owing to the faster degradation of Sr-BGs versus undoped BGs, which can arise from the substitution of Sr with a bigger cation radius (1.16 Å) for smaller Ca (0.94 Å) and expansion of the Sr-BG network. Based on these considerations, the network becomes weakened in the state of doping with Sr and accelerates the degradation rate of scaffolds [15,19,41].

### 3.2. Biological Evaluation Findings

#### 3.2.1. Osteoactivity Profile

Figure 8 illustrates the SEM of the lyophilized hydrogels containing bioglass (undoped and Sr-doped) after soaking in SBF for seven days. A notable property of bioactive scaffolds is their potential to attach to the bone via hydroxyapatite formation on their surface. As is evident in the images, the scaffolds demonstrated full coverage of the surface with apatite deposits, though their mineralization amount was diverse. Visual inspection of the Alg/Sr-BG scaffolds followed by submersion in SBF reveals that sediment of mineral phase on the surface scaffolds increased with the Sr-BG concentration’s incremental increase, likely due to the availability of more nucleation sites, surface charges, and mineralization [42,43].

#### 3.2.2. Hemocompatibility Results

The hemolysis percentages indicate the extent to which hemoglobin was released into the plasma solution from the broken red erythrocytes after exposure to the tested substances [36,44,45]. A lower hemolysis degree for a scaffold means fewer lysed blood cells and, therefore, greater blood compatibility. In the current work, the hemolytic potentials of the treated alginate-based composites were compared with each other and with negative and positive control groups, distilled water, and PBS, respectively. According to Figure 9, all four experimental samples resulted in a hemolysis % lower than 5%, agreeing with the permissible international limit [15,44,46,47]. However, it was observed that erythrocytes have a lesser % hemolysis when exposed to doses of superior Sr-BG. The difference between the hemolysis percentages of Alg/BG, Alg/Sr-BG 5%, and Alg/Sr-BG 10% was not statistically significant. Consequently, alginate hydrogels containing Sr-substituted BG show high adaptability with blood cells.

#### 3.2.3. Cell Viability Findings

The effect of different formulations of developed substrates on the cellular viability of osteoblast-like MG-63 cells after 24 and 72 h was evaluated using an MTT assay (Figure 10). According to the acquired data from the MTT assay in Figure 10, none of the nanocomposite scaffolds exhibited cytotoxicity on the model cell line, and since all showed viability above 70%, they can be considered biocompatible materials as defined by the ISO 10993–5:2009 [48,49]. It is also clear in Figure 10 that the osteoblast-like cells, in all groups, showed an increase in viability percentage with increasing incubation time from 1 to 3 days, denoting the cell attachment and proliferation on the nanocomposite scaffolds. The obtained results show that the highest cell proliferation occurs when exposed to a Alg/Sr-BG 15% scaffold at 72 h, so this was statistically significant in comparison with the other groups, except for Alg/Sr-BG 10%. Furthermore, at 72 h, the cell growth of examination groups was superior to the control group, excluding Alg, which is less than the control.

## 4. Discussion

BGs can be considered for hard tissue regeneration due to possessing multiple bioactivities and excellent osteoconductivity ability. Their ability to release vital ions such as silica and calcium in the biological environment promotes the possibility of angiogenesis and bone formation in the body. Therefore, by replacing some of their ions with medical metal ions such as copper or zinc or strontium, their biological properties can be improved [13,15,16,17]. In recent research, BG incorporation in alginate hydrogels was studied and applied for wound healing and bone scaffold [13,32]. Our research path in this work was concentrated on the fabrication and evaluation of alginate composite scaffolds reinforced with Sr-doped bioglass. In the initial preparation, Sr-BG nanofibers were produced using an electrospinning method combined with sol–gel, which turned into pure Sr-BG nanoparticles after a 600 °C temperature treatment. The SEM images of Sr-BG in Figure 1 show that the diameter size of nanofibers decreases after firing due to the removal of organic polymer. Examining the chemical composition of nanofibers before and after firing shows the same presence of Sr ions in both cases (Figure 1).

After uniformly dispersing the powdered Sr-BG in the Alg matrix, CaCl_2_ solution was added to the constructor to crosslink the matrix. Before and after gelling (with Alg), the structural characteristics of the composite scaffold were examined by SEM-EDS, FTIR, and XRD.

Optimized pore size, interconnected structure, rough surface, and higher surface area are significant factors in the design of the next generation of scaffolds for bone tissue engineering. It has been reported that pore size in the range from 75 μm to 250 μm is the most suitable pore size supporting vascularization and transport of gases and nutrition/wastes as well as enabling migration, infiltration, and adhesion of cells. Conforming to the SEM images, optimized pore size can be seen in the porous structure of strontium-containing hydrogel scaffolds, which is in the appropriate pore size range for bone tissue scaffolds. Examining the FTIR and XRD peaks in Figure 4 and Figure 5 and comparing the results with previous reports confirmed the successful incorporation of the bioglass into alginate hydrogel and the amorphous nature of the hydrogel composites.

Changes in the swelling ratio of Sr-BG-containing hydrogels reached saturation and flattened after a longer period of time compared to the BG-containing samples (Figure 6), which could be due to the smaller pore size of hydrogels in the presence of Sr-BG. Furthermore, according to Figure 6, the swelling rate decreases with an increasing Sr-BG value, in line with a reduction in pore size.

As seen in Figure 7, degradation of Alg/Sr-BG in an acidic environment mimicking the environment of bone defects was quicker than that in PBS medium, which can increase the concentration of Sr ions at the lesion site. This phenomenon can be useful for bone homeostasis based on the positive effect of strontium on osteoblast differentiation and inhibition of osteoclastogenesis [18].

Based on previous reports by other researchers, the pure alginate scaffolds do not provide significant bioactivity [32,50]. The inclusion of the BG nanoparticles into the alginate matrix was shown to greatly enhance the in vitro mineralization on the surface of the nanocomposites under a simulated physiological medium [32,38,50]. Likewise, Özge Gönen et al. [51] found that in vitro osteoactivity of gelatin/poly(e-caprolactone) (Gt/PCL) nanofiber mats significantly increased when reinforced with Sr-BG. When the osteoactivity of the scaffolds was studied in SBF solution, the mineralization on the surface of the nanocomposites occurred to a significant extent [32,50]. Substituting Sr for Ca in BG enhanced apatite phase constitution on the surface of Ca-crosslinked Alg frameworks when immersed in SBF solution.

Various studies have shown that BG-containing scaffolds have accelerated cell proliferation and good cell adherence [52,53]. For instance, Jayakumar et al. [50] found that the embedment of BG particles into the Alg hydrogels matrix did not adversely affect the viability and compatibility of MG-63 cells and induced cell attachment and proliferation onto the scaffolds. Our in vitro evaluations exhibited a significant cell proliferation rate on the synthesized composites, especially for Alg/Sr-BG 10 and 15%, as well as appropriate blood compatibility. The results indicate that the fabricated hydrogel nanocomposite possesses beneficial structural, physical, and biological properties and can be a promising scaffold for bone tissue engineering applications. However, the bone fracture healing potential of the fabricated scaffold must be evaluated in a proper animal model to facilitate its clinical translation.

## 5. Conclusions

In general, Alg/Sr-BG composite scaffolds were successfully fabricated using techniques of electrospinning integrated with sol–gel, ion crosslinking, then lyophilization, and characterized. It was found that the composite scaffolds had a pore size of about 230–330 nm, sufficient porosity, controlled swelling capacity, acceptable degradation rate, and enhanced bioactivity, all of which were influenced by the amount of bioglass incorporated. Incorporating Sr-BG significantly enhanced the viability of MG-63 cells and decreased blood cell hemolysis. All these results suggested that an Alg/Sr-BG composite scaffold could serve as an appropriate bioactive matrix for bone tissue regeneration.

## Figures and Tables

**Figure 1 biology-11-01472-f001:**
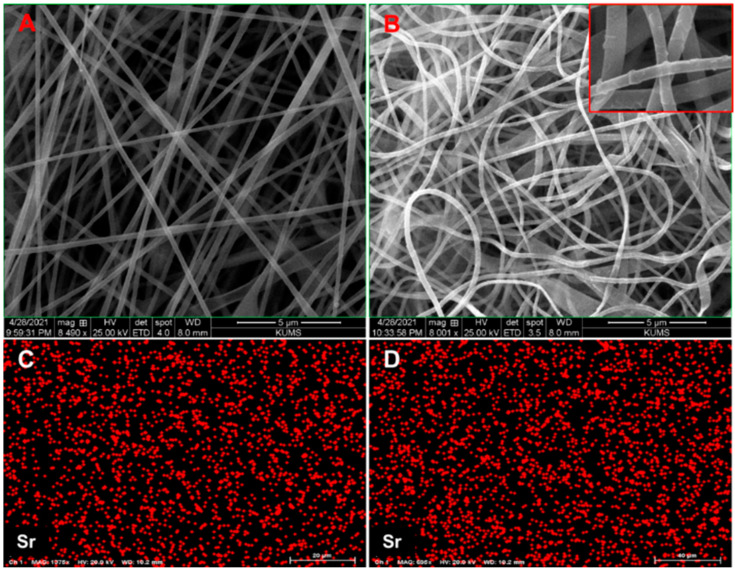
The SEM images of (**A**) PVA/Sr-BG nanofibers (**B**) calcined PVA/Sr-BG nanofibers, and EDX analysis of (**C**) PVA/Sr-BG nanofibers (**D**) calcined PVA/Sr-BG nanofibers, indicating Sr.

**Figure 2 biology-11-01472-f002:**
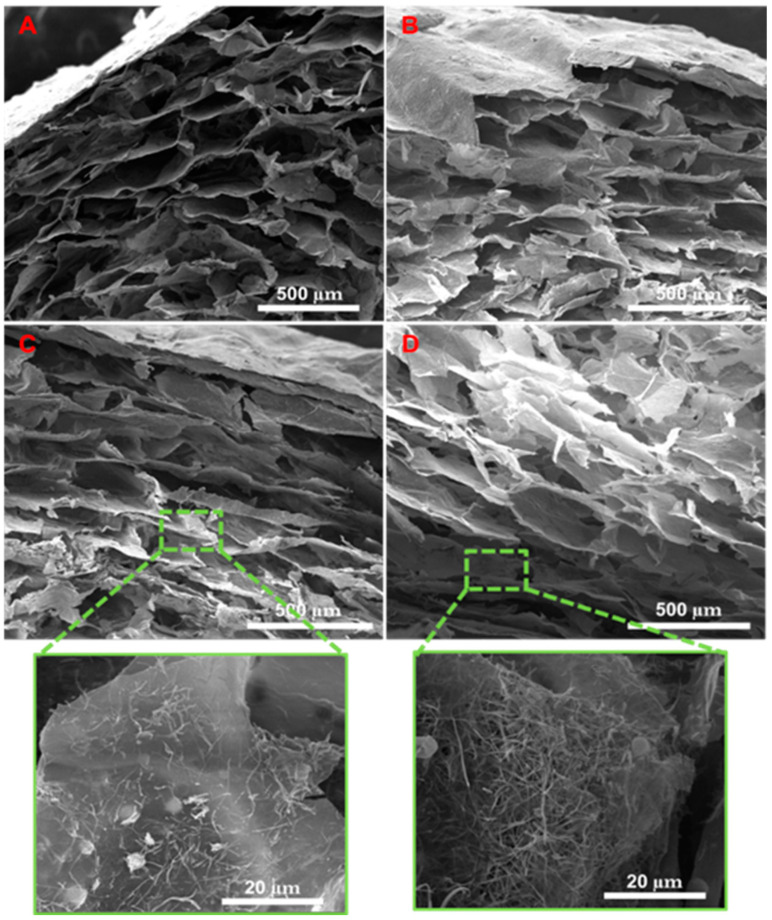
Scanning electron micrographs of the 3D nanocomposite scaffolds. (**A**) Alg/BG 10%, (**B**) Alg/Sr-BG 5%, (**C**) Alg/Sr-BG 10%, and (**D**) Alg/Sr-BG 15%.

**Figure 3 biology-11-01472-f003:**
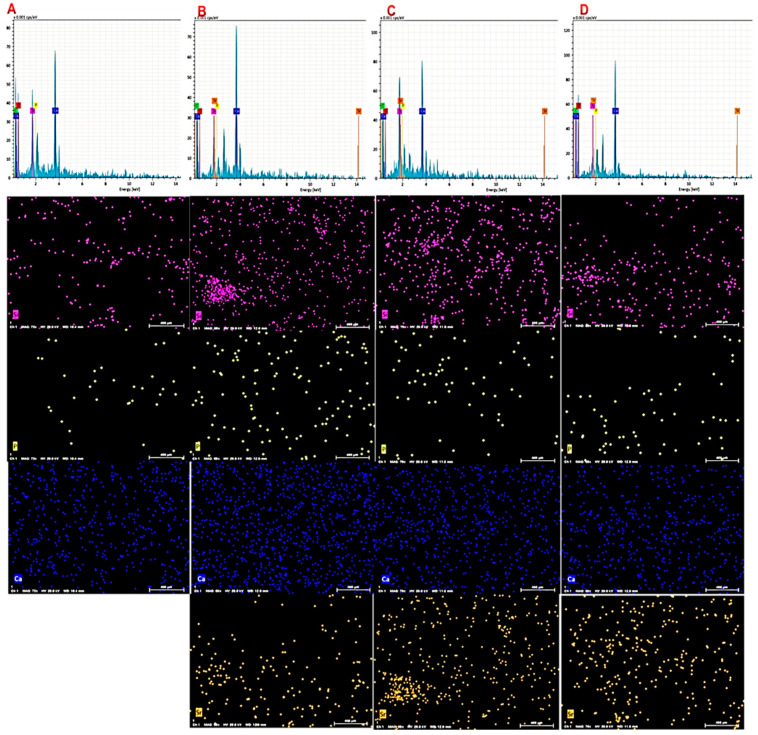
EDS mapping of Si (purple), P (yellow), Ca (blue), and Sr (orange) of prepared scaffold: (**A**) Alg/BG 10%, (**B**) Alg/Sr-BG 5%, (**C**) Alg/Sr-BG 10%, and (**D**) Alg/Sr-BG 15%.

**Figure 4 biology-11-01472-f004:**
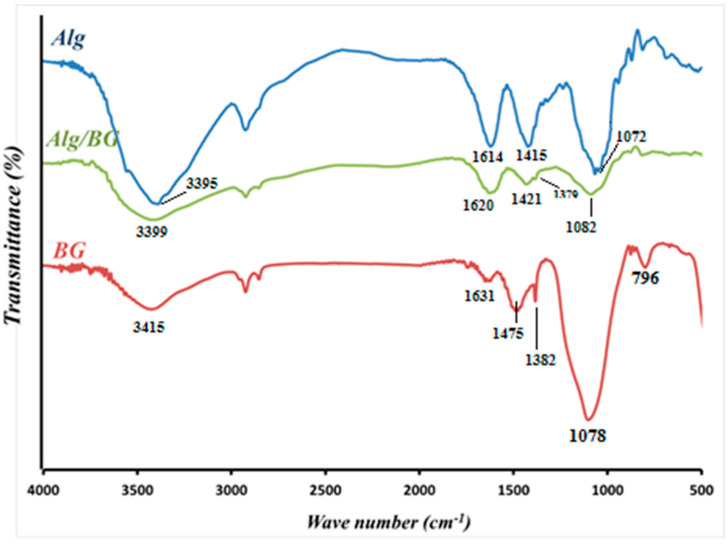
FTIR spectra of Alg (blue), BG (red), and Alg/BG hydrogel (green).

**Figure 5 biology-11-01472-f005:**
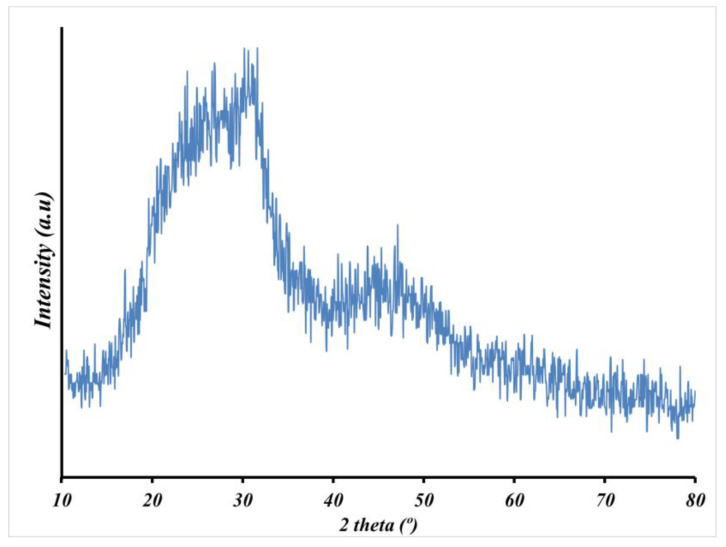
XRD spectra of the sintered Sr-BG nanofibers at 600 °C for 5 h.

**Figure 6 biology-11-01472-f006:**
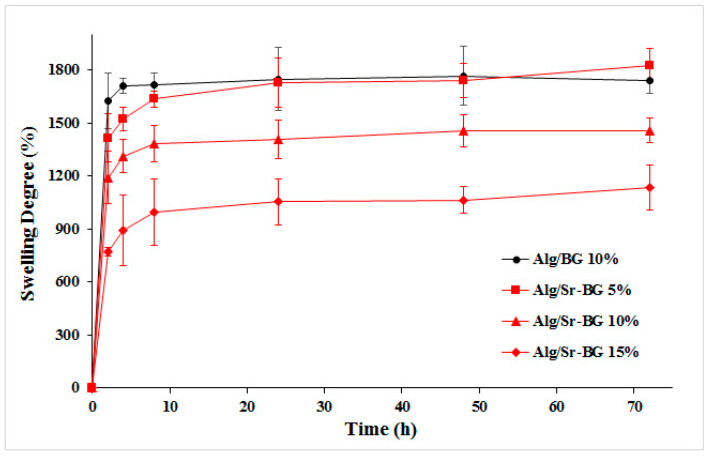
The trend of the swelling degree (%) of the Alg/BG and Alg/Sr-BG (5%, 10% and 15%) hydrogels’ time dependence over a period of 72 h.

**Figure 7 biology-11-01472-f007:**
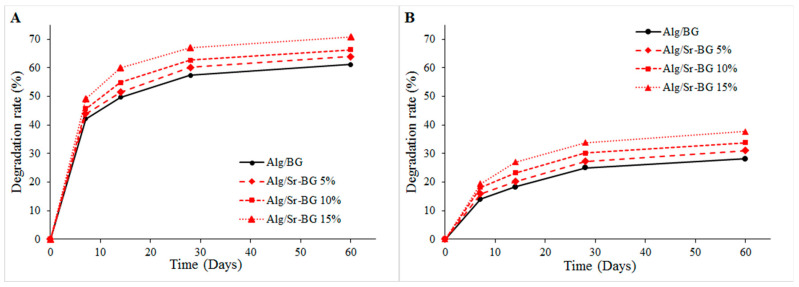
Biodegradation profile of four kinds of alginate scaffolds as a function of immersion time in (**A**) AcOH buffer and (**B**) PBS over 60 days.

**Figure 8 biology-11-01472-f008:**
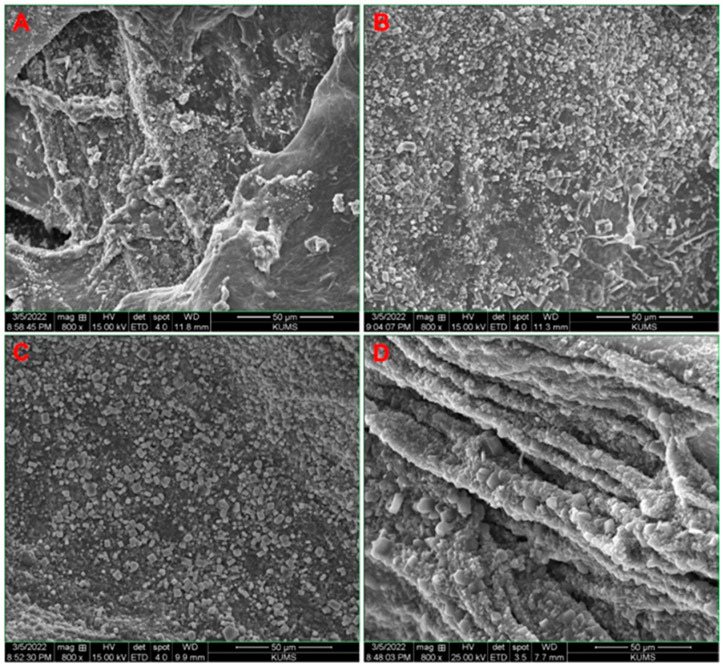
SEM graphs of SBF-submerged substrates showing apatite nanocrystals deposits on (**A**) Alg/BG, (**B**) Alg/Sr-BG 5%, (**C**) Alg/Sr-BG 10%, and (**D**) Alg/Sr-BG 15% after soaking in SBF for 7 days.

**Figure 9 biology-11-01472-f009:**
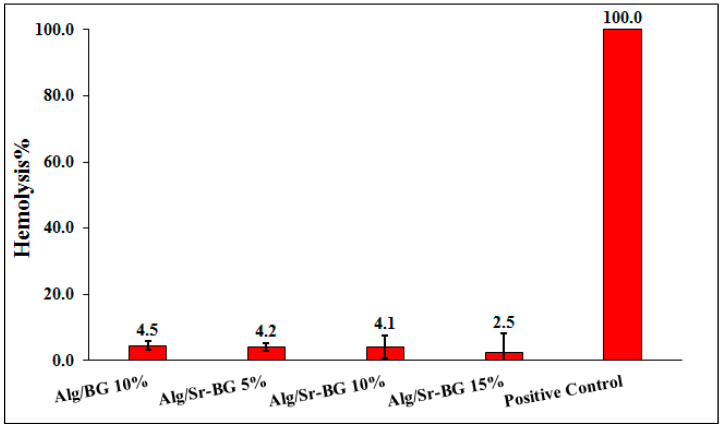
Hemolysis percentage of the prepared gel composites compared to positive control.

**Figure 10 biology-11-01472-f010:**
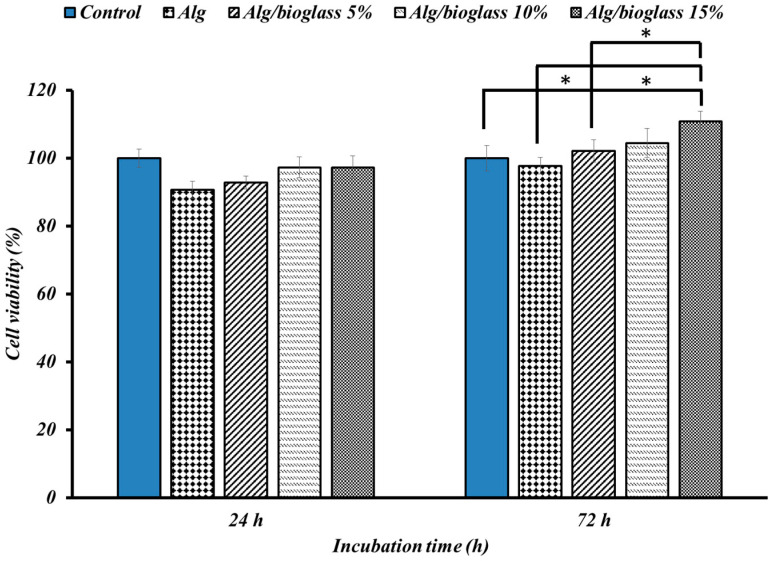
Cell proliferation as determined by MTT assay of MG-63 cells cultured in bioactive glass for 24 and 72 h. *: *p* < 0.05.

## Data Availability

Further data are available on reasonable request from the corresponding author.

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
