# Peer review of "Nanofibrous Hydrogel Nanocomposite Based on Strontium-Doped Bioglass Nanofibers for Bone Tissue Engineering Applications"

_biology, 2022, doi:10.3390/biology11101472_

Round 1

Reviewer 1 Report

This paper presents a nanofibrous hydrogel nanocomposite based on strontium-2 doped bioglass nanofibers. The properties of the hydrogel were well studied and presented. However, the paper lacks the bone tissue engineering related data to support its title and abstract. Lots of experiments should be added if the authors claim their application is bone tissue engineering.

Author Response

Review Report 1

Comments

This paper presents a nanofibrous hydrogel nanocomposite based on strontium-2 doped bioglass nanofibers. The properties of the hydrogel were well studied and presented. However, the paper lacks the bone tissue engineering related data to support its title and abstract. Lots of experiments should be added if the authors claim their application is bone tissue engineering.

Response: We greatly appreciate your suggestion and guidance. We greatly appreciate your suggestion and guidance. In this study, an attempt has been made to examine and compare the most common methods and analyzes, as well as suitable biological materials that can be used to construct bone scaffolds. Considering the limitations of laboratory studies, our research showed that this scaffold has favorable bioactivity, biocompatibility, and biodegradability properties and allows for more tests and analyses in our research group's next research.

Reviewer 2 Report

  The authors submitted a manuscript investigating the role of an alginate-based hydrogel reinforced with Sr-doped bioglass nanocomposites in the application of bone tissue engineering.

  With the development of bone tissue engineering, bio-scaffold materials by adding metallic ions to improve bone healing have been extensively explored in the past decades. Sr is a trace element taken with nutrition and found in bone in close connection to native hydroxyapatite. Sr is involved in a dual mechanism of coupling the stimulation of bone formation with the inhibition of bone resorption.

  The manuscript’s perspective is somewhat innovative for the application of nanocomposite hydrogels. Overall, this manuscript is better designed, executed, written, and the logic is relatively clear.

  Nonetheless, there are a number of issues that need to be noted.

  1. Hydrogels have been widely used in bone tissue engineering research, but there is a problem of poor mechanical properties, and improving the mechanical properties of hydrogels can improve their bone repair effect. The combination of nanoparticles and hydrogels can improve the mechanical properties of hydrogels. Therefore, what are the mechanical properties of the scaffolds prepared by the authors?

  2. Based on the above point, the authors should clearly discuss the future direction of the scaffold for specific applications, rather than generalizing it for bone tissue engineering.

  3. The English needs to be improved to a certain extent. There are some errors in grammar and format in the whole manuscript: inconsistencies; single and plural expressions; the use of prepositions and definite/indefinite articles; uppercase and lowercase; the abbreviations; superscript and subscript; punctuation.

Author Response

Review Report 2

Comments and Suggestions for Authors

  The authors submitted a manuscript investigating the role of an alginate-based hydrogel reinforced with Sr-doped bioglass nanocomposites in the application of bone tissue engineering.

  With the development of bone tissue engineering, bio-scaffold materials by adding metallic ions to improve bone healing have been extensively explored in the past decades. Sr is a trace element taken with nutrition and found in bone in close connection to native hydroxyapatite. Sr is involved in a dual mechanism of coupling the stimulation of bone formation with the inhibition of bone resorption.

  The manuscript’s perspective is somewhat innovative for the application of nanocomposite hydrogels. Overall, this manuscript is better designed, executed, written,

  Nonetheless, there are a number of issues that need to be noted.

  1. Hydrogels have been widely used in bone tissue engineering research, but there is a problem of poor mechanical properties, and improving the mechanical properties of hydrogels can improve their bone repair effect. The combination of nanoparticles and hydrogels can improve the mechanical properties of hydrogels. Therefore, what are the mechanical properties of the scaffolds prepared by the authors?

Response: Thanks for your valuable comment. Unfortunately, due to some instrumental and financial shortcomings, we cannot conduct the mechanical properties evaluations of the fabricated hydrogels. Moreover, several hydrogel-based scaffolds with poor mechanical properties have been evaluated as bone tissue engineering scaffolds. These scaffolds act as a substrate for cell proliferation and migration and must be mechanically protected in load-bearing bones or applied in non-load-bearing scaffolds.

  1. Based on the above point, the authors should clearly discuss the future direction of the scaffold for specific applications, rather than generalizing it for bone tissue engineering.

Thank you for your guidance. Several studies have studied and proven the potential use of alginate scaffolds containing bioglass in the reconstruction (or replacement) of soft tissue such as skin and cartilage and hard tissue such as cortical bone. Therefore, our Alg/Sr-BG scaffold can be studied and investigated not only for application in soft tissue engineering, such as wound healing but also for any disease that simultaneously contains hard and soft tissue.

  1. The English needs to be improved to a certain extent. There are some errors in grammar and format in the whole manuscript: inconsistencies; single and plural expressions; the use of prepositions and definite/indefinite articles; uppercase and lowercase; the abbreviations; superscript and subscript; punctuation.

Response: Thank you for in-deep reviewing the manuscript. The manuscript was rechecked and the mistakes were corrected.

Reviewer 3 Report

-Conclusions are missing

-Figure 3 should be improved, it is not well appreciated

- It is not clear if they have the permits for ethical tissue handling.

- The XRD spectrum is incomplete, does not have the PDFs, and the peaks are not indexed (the peaks do not have the miller indices), nor do they show any calculation of the basic crystal parameters. There is no interpretation of where the metals are located in the crystal structure.

- The infrared spectra do not show the bonds, nor do they specify what the curves are of or in what ranges, nor do they explain their occurrence.

- It must be proved that the nanofibers were doped with strontium,

-It is recommended to read the following manuscripts:

Doped Potassium Jarosite: Synthesis, Characterization and Evaluation as Biomaterial for Its Application in Bone Tissue Engineering. Metals, 12, 1052. 2022

Cytotoxicity and Cell Growth Assays,  In Cell biology 2006

Author Response

Review Report 3

Comments and Suggestions for Authors

1- Conclusions are missing

Response: Thanks for your careful comment. We corrected this point in the revised manuscript.

2- Figure 3 should be improved, it is not well appreciated

Response: We respect your comment. As recommended, the quality of figure 3 was improved in the revised manuscript, and more explanations were given about it.

3- It is not clear if they have the permits for ethical tissue handling.

Response: Thanks for your comment, the code of ethics is presented in the "Funding" section under the title "Ethics No".

4- The XRD spectrum is incomplete, does not have the PDFs, and the peaks are not indexed (the peaks do not have the miller indices), nor do they show any calculation of the basic crystal parameters. There is no interpretation of where the metals are located in the crystal structure.

Response: Thank you for your insightful comment. As indicated in the manuscript, the synthesized nanofibers exhibited amorphous nature and lack of crystalline.

5- The infrared spectra do not show the bonds, nor do they specify what the curves are of or in what ranges, nor do they explain their occurrence.

Response: Thank you for your guidance. For better comprehension of FTIR spectra, in the modified article, wavenumbers related to the functional groups of each compound were included in the corresponding diagram.

6- It must be proved that the nanofibers were doped with strontium,

Response: Thanks for your suggestion. Figures 1C and 1D show the presence and, with high probability, strontium doping into the nanofibers. 

7- It is recommended to read the following manuscripts:

Doped Potassium Jarosite: Synthesis, Characterization and Evaluation as Biomaterial for Its Application in Bone Tissue Engineering. Metals, 12, 1052. 2022

Cytotoxicity and Cell Growth Assays,  In Cell biology 2006

Response: Thank you for in-deep reviewing the manuscript and your excellent recommends.

Reviewer 4 Report

It was a manuscript about the synthesis and evaluation of Alg/Sr-BG nanocomposite scaffold for the aim of bone tissue engineering applications. Here are some comments for this study that should be considered before publication:

1-    The quality of the abstract needs to be improved.

2-    “Previous research has proven the ability of calcium alginate gel to be used as a scaffold or cell carrier for bone reconstruction” please add some references related to this sentence.

3-    Please use the same phrase for the fabricated scaffolds in the whole text. In some cases, you used scaffold and in some place sponge.

4-    Please use the same format for referencing in the main text.

5-    Please add the color-related element in the figure caption of figure 2.

6-    Please show the main FTIR peaks in the figure as well.

7-    Please add the XRD results of BG, Sr-BG, Alg/Sr-BG, and Alg/BG, as well.

8-    Is the degradation rate of Sr-contained scaffold in the safe window or it could lead to undesirable tissue defects? Indeed, according to Figure 7, the scaffolds showed a high-speed degradation rate in low pH, is this good?

9-    “It has been reported that the pore size in the range from 75 μm to 250 μm is the most suitable pore size supporting vascularization and transport of gases and nutrition/wastes as well as enabling migration, infiltration, and adhesion of cells.” please add references related to this sentence.

10- “. Conforming 446 to the SEM images, optimized pore size, can be seen in the porous structure of strontium-containing hydrogel scaffolds, which is in the appropriate pore size range for bone tissue scaffolds. [25-27]. Examining the FTIR and XRD peaks in Figures 4 and 5 and comparing the results with previous reports confirmed the successful incorporation of the bioglass in alginate hydrogel and the amorphous nature of the hydrogel composites [29-31].” these are your results, so references are not needed for them.

11- The discussion part needs to be improved.

12- Please separate the conclusion section from the discussion and mention it under a new subheading. Moreover, please improve its quality.

13- Please compare the swelling degree of different Alg/Sr-BG samples.

14- Why did you choose MG-63 cells for the MTT test?

15- In conclusion, the results indicate that the fabricated hydrogel nanocomposite possesses beneficial structural, physical, and biological properties and can be a promising scaffold for bone tissue engineering applications.” also these tests are not sufficient for confirming the capability of the suggested scaffold for bone tissue engineering, and other tests like alkaline phosphatase, calcium assay, and Alizarin Red staining are needed as well. 

Author Response

Review Report 4

Comments and Suggestions for Authors

It was a manuscript about the synthesis and evaluation of Alg/Sr-BG nanocomposite scaffold for the aim of bone tissue engineering applications. Here are some comments for this study that should be considered before publication:

1- The quality of the abstract needs to be improved.

Response: We greatly appreciate the reviewer's valuable opinion. In the revised version of the article, corrections were made to improve the quality of the abstract.

2- “Previous research has proven the ability of calcium alginate gel to be used as a scaffold or cell carrier for bone reconstruction” please add some references related to this sentence.

Response: Thanks for your suggestion. The recommended point was included in the revised version.

3- Please use the same phrase for the fabricated scaffolds in the whole text. In some cases, you used scaffold and in some place sponge.

Response: Thank you for your insightful comment. The manuscript was rechecked and your suggested corrections have been made.

4- Please use the same format for referencing in the main text.

Response: Thanks for your careful comment. We have discussed this point in the revised manuscript.

5- Please add the color-related element in the figure caption of figure 2.

Response: Thank you for the constructive comment. We assumed you meant figure 3, and we put the corresponding color of each element in the EDS graph in the figure description section.

6- Please show the main FTIR peaks in the figure as well.

Response: Thank you for your guidance. For better comprehension of FTIR spectra, in the modified article, wavenumbers related to the functional groups of each compound were included in the corresponding diagram.

7- Please add the XRD results of BG, Sr-BG, Alg/Sr-BG, and Alg/BG, as well.

Response: Thank you for your constructive comment. Unfortunately, due to some financial shortcomings, we cannot conduct the requested analysis. 

8- Is the degradation rate of Sr-contained scaffold in the safe window or it could lead to undesirable tissue defects? Indeed, according to Figure 7, the scaffolds showed a high-speed degradation rate in low pH, is this good?

Response: Thank you for your detailed review. The degradation rate of a bone scaffold must be lower than the speed of restoration of the damaged bone to maintain the integrity of the scaffold for cell adhesion and proliferation; on the other hand, the pores created in effect degradation activate the deposition of hydroxyapatite on the scaffold and accelerate the growth of apatite. It also provides the necessary space for the formation of blood vessels and new tissue and thus bone repair.

We have shown that by increasing the amount of Sr-BG, the rate of degradation increases in both acidic and neutral pH. The desired degradation rate can be adjusted by using the appropriate amount of incorporated Sr-BG. The acidic environment is the physiological environment around the injured bone that should be considered in in vivo experiments.

9- “It has been reported that the pore size in the range from 75 μm to 250 μm is the most suitable pore size supporting vascularization and transport of gases and nutrition/wastes as well as enabling migration, infiltration, and adhesion of cells.” please add references related to this sentence.

Response: Thanks for your careful comment.

  1. Biosilica incorporated 3D porous scaffolds for bone tissue engineering applications, Materials Science & Engineering C, 91 (2018) 274–291.
  2. Bioengineered 3D nanocomposite based on gold nanoparticles and gelatin nanofibers for bone regeneration: in vitro and in vivo study, Scientific Reports, 11 (2021) 13877–13888
  3. A critical review on polymer-based bio-engineered materials for scaffold development, Composites: Part B 38 (2007) 291–300.

10- “. Conforming 446 to the SEM images, optimized pore size, can be seen in the porous structure of strontium-containing hydrogel scaffolds, which is in the appropriate pore size range for bone tissue scaffolds. [25-27]. Examining the FTIR and XRD peaks in Figures 4 and 5 and comparing the results with previous reports confirmed the successful incorporation of the bioglass in alginate hydrogel and the amorphous nature of the hydrogel composites [29-31].” these are your results, so references are not needed for them.

Response: Thank you for your guidance. Necessary corrections were applied in the revised manuscript.

11- The discussion part needs to be improved.

Response: Thank you for your guidance. The discussion section was qualitatively modified in the revised manuscript.

12- Please separate the conclusion section from the discussion and mention it under a new subheading. Moreover, please improve its quality.

Response: Thanks for your careful comment. We rewrote the conclusion section in a separate part of the revised manuscript.

13- Please compare the swelling degree of different Alg/Sr-BG samples.

Response: Thanks for your valuable comment. The proposed subject was included in the revised manuscript's description of the swelling rate measurement.

14- Why did you choose MG-63 cells for the MTT test?

Response: Thank you for your insightful comment. Usually, osteoblast-like cell line MG-63, as a prototype of bone cells, is used for bone tissue engineering cell studies.

15- “In conclusion, the results indicate that the fabricated hydrogel nanocomposite possesses beneficial structural, physical, and biological properties and can be a promising scaffold for bone tissue engineering applications.” also these tests are not sufficient for confirming the capability of the suggested scaffold for bone tissue engineering, and other tests like alkaline phosphatase, calcium assay, and Alizarin Red staining are needed as well.

Response: Thank you for in-deep reviewing the manuscript. We completely agree with the honorable reviewer's suggestion.

Round 2

Reviewer 1 Report

.

Reviewer 4 Report

Thanks for addressing the comments.